# A Study of Blood Pressure and Physical Fitness in People with Intellectual Disabilities in South Korea

**DOI:** 10.3390/healthcare12090887

**Published:** 2024-04-25

**Authors:** Bogja Jeoung

**Affiliations:** Department of Exercise Rehabilitation, Gachon University, 191 Hambakmeo-ro, Yeonsu-gu, Incheon 21936, Republic of Korea; bogja05@gachon.ac.kr

**Keywords:** physical fitness, intellectual disabilities, blood pressure, body composition

## Abstract

Individuals with intellectual disabilities have a shorter lifespan and significantly higher prevalence of conditions such as hypertension and cardiovascular diseases than healthy individuals. Thus, assessing the elements that contribute to their physical fitness is crucial. This cross-sectional study examined the relationship between the blood pressure and physical fitness of people with intellectual disabilities in South Korea, considering differences across sexes, age groups, physical attributes, and disability levels. It used data from 8502 individuals with intellectual disabilities aged 20–59 years who participated in a survey of a National Fitness Standard Center (NFSC) between 2018 and 2021. A series of *t*-tests, one-way analysis of variance, logistic regression, and the four-quartile method were used for data analyses. The results showed differences in physical fitness levels between men and women considering all aspects except for BMI (Body Mass Index), with men showing higher blood pressure levels. Lower grip strength, lower PEI (physical efficiency index) scores, and higher BMI were associated with increased blood pressure. Additionally, individuals with higher levels of disability tended to have lower levels of physical fitness, while higher physical fitness levels were associated with lower blood pressure. Therefore, low fitness levels and hypertension risk may be important health indicators for people with intellectual disabilities.

## 1. Introduction

The term “vulnerable groups” generally refers to people who, compared to others, have relatively limited opportunities for social participation due to economic, physical, and other conditions and are likely to be excluded from the opportunity to receive equal benefits as members of society without national intervention. These include people with disabilities, who have considerably low access to health care and high-risk factors, thus necessitating national-level health care.

Intellectual disabilities refer to conditions characterized by significant limitations in intellectual functioning and adaptive behavior from before 22 years old [1]. According to the Enforcement Rules of the Act-On Welfare of Persons with Disabilities, intellectual disabilities are classified as mild (Level 3) based on an intelligence quotient (IQ) and social quotient (SQ) of 50–70 or less; moderate (Level 2) based on an IQ and SQ of 35–49; and severe or profound (Level 1) based on an IQ and SQ of 34 or less [2].

In South Korea, the number of people with intellectual disabilities has continuously increased [3]. According to 2022 data, 75%, 19%, and 5% of people with intellectual disabilities were aged 6–17, 18–64, and 65 years and older, respectively. Children with intellectual disabilities have poorer motor skills and lower physical fitness levels than children without intellectual disabilities, typically causing the former to have a shorter lifespan and age faster [4]. According to research, the aging process of individuals with intellectual disabilities begins around the ages of 40–50 years [5]. Similarly, adults with intellectual disabilities have reduced physical fitness and higher morbidity from chronic diseases such as obesity, hyperlipidemia, hypertension, and diabetes compared to those without intellectual disabilities, leading to an early mortality rate and low quality of life for the former [6]. These health problems are caused by several factors, such as having a sedentary lifestyle with reduced physical activity. Therefore, to assist this population in maintaining their physical function and improving their health and quality of life, research is needed to assess their physical fitness and understand the causes of the decline in physical function [7].

Among the above-mentioned chronic conditions, hypertension is a cardiovascular disease (CVD) risk factor for these individuals—as for the general population—and a major cause of death [8]. Individuals with intellectual disabilities show a higher prevalence of the two major hypertension risk factors: obesity and low physical activity [9,10]. Therefore, the significance of physical fitness levels for individuals with intellectual disabilities, who often lead sedentary lifestyles, is emphasized for their health.

Numerous studies have indicated that individuals with intellectual disabilities engage in less physical activity compared to healthy individuals, leading to a higher prevalence of metabolic syndrome, including hypertension and diabetes. These studies have particularly emphasized the relationship between physical fitness and health outcomes. Among the components of physical fitness, cardiorespiratory endurance, muscular strength and endurance, and obesity have been identified as factors contributing to the increased risk of hypertension [11,12,13].

Participating in physical activities improves physical fitness. High physical fitness is a strong predictor of lowered morbidity and mortality risks associated with metabolic syndrome, type 2 diabetes, and CVDs [14,15]. Increases in physical activity and strength decrease the risk of CVDs by approximately 25% and 60%, respectively. While the risk of CVDs related to physical activity decreases gradually, physical fitness reaching the bottom 25% can reduce the relative risk of CVDs by 40%. Hence, physical fitness contributes more to health than physical activity [16]. These findings confirm that prior research has established a correlation between health-related physical fitness and CVD in individuals with intellectual disabilities.

In South Korea, the national physical fitness certification system for individuals with disabilities was introduced to systematically manage their physical fitness. Consequently, national fitness standard centers (NFSCs) of the Korea Paralympic Committee (KPC) have been established. Fitness test parameters for people with disabilities were developed considering several conditions, including relevance to each type of disability, easy performance without the risk of injury, and scientific standards with good reliability and validity [17].

Research data systematizing the physical fitness indicators of individuals with intellectual disabilities is extremely scarce, and there is a severe lack of studies that compare and analyze their blood pressure—a cardiovascular risk factor—to assess its effects. Therefore, this study examined the relationship between physical fitness and blood pressure risk, considering their risk factors, based on data from the NFSCs from 2018 to 2021.

## 2. Materials and Methods

### 2.1. Data Collection

This study utilized a cross-sectional design. Data were obtained from the Korea Culture Information Sports Association’s big data market, which is open to the public. Twelve of the Korea Paralympic Committee (KPC) fitness standard test centers released four years of data (2018–2021) during data collection. This study examined 8,502 participants with intellectual disabilities (*N* = 8230; 96.8%) and autism (*N* = 272; 3.2%). These participants were classified according to different levels of disability: Levels 1 to 3, as shown in Table 1. The ages of the participants ranged between 20 and 59 years old, of which 61.0% were men and 39% were women (see Table 1 for detailed participant characteristics). Informed consent was obtained from all participants or their legal representatives. We used STROBE checklist to ensure transparency and completeness (see Appendix A). 

### 2.2. Physical Fitness and BMI Measurement

The health-related physical fitness tests developed for people with disabilities comprise five parameters: (a) cardiovascular endurance, (b) upper-body muscle strength, (c) muscle endurance, (d) flexibility, and (e) body compositions [17,18]. This study measured all the physical fitness ((a) to (d)) and body composition (e) parameters, which were measured at the Fitness Standard for People with Disabilities Center. The BMI was calculated by dividing body weight (kg) by height in square meters (m^2^). Each physical fitness test showed high internal consistency with satisfactory reliability statistics (Cronbach’s alpha) ranging from α = 0.70 to 0.93 [17,18]. All items were assessed by instructors with national professional health and fitness certificates.

First, cardiovascular endurance was measured through a 3-min step test. Using a step box (30 cm height for men and 20 cm height for women), the step test was performed at a beat rate of 30 and 24 steps per min for men and women, respectively. After the test, the participants were instructed to sit, and their heart rates were measured for 30 s each across three different time points: between 1 and 1.5 min, 2 and 2.5 min, and 3 and 3.5 min. The measured heart rate was used to calculate the physical efficiency index (PEI). The PEIs of participants aged 8–88 years were calculated with the equation PEI = (D × 100)/(2 × P), whereas those of women aged ≥ 18 years and men aged >16 years were calculated as PEI = 0.22 × (300 − D) + (D × 100)/(5.5 × P), where “D” indicates the step test duration (i.e., 180 s), and “P” represents the summed pulse counts across the three points.

Controversy remains over the method of measuring cardiovascular endurance because it is challenging for individuals with intellectual disabilities to step in line with a metronome owing to their lack of cognitive understanding. Furthermore, cardiovascular endurance test methods for individuals with intellectual disabilities vary across studies. Winnick and Short [19] used the PACER (Progressive Aerobic Cardiovascular Endurance Run) method with a distance of either 20 m or 16 m. Jeon and Han [20] utilized the PEI with a 3-min step test. However, Chow et al. [21] and Wouters et al. [22] employed a 6-min walk test to measure cardiovascular endurance. The National Fitness Standard Charts (NFSCs) provided by the KPC offer two options for cardiovascular endurance testing for individuals with intellectual disabilities: a 6-min walk test or a 3-min step test. The choice of test method may depend on various factors, including the study’s objectives and the participants’ capabilities. In this study, considering the situation of the places where the tests were conducted (the tests were conducted on-site during client visits or when traveling to the sites), a 3-min step test was employed, following the national fitness standard of the Korean Physical Fitness Certification (KPC). Lee [17] also selected this test as the final measurement item for fitness certification for individuals with intellectual disabilities.

Second, a hand-grip strength test was used to measure upper-body muscle strength. This test was performed twice for the left and right hands using a hand dynamometer, and the highest value was recorded to the nearest 0.1 kg. Third, a sit-up test was performed to estimate the endurance strength of the abdominal muscles and hip flexors. Participants were asked to touch their knees with their elbows, return to the mat, and continue to perform as many repetitions as possible in 60 s. Finally, the sit-and-reach test was used to measure body flexibility, which measures the range of motion of the spine and hip during deep trunk flexion. The participants were instructed to sit barefoot with legs extended, toes pointed up, feet approximately hip-width apart, and the soles of their feet against the base of the measuring device. Subsequently, they were asked to slowly push and slide forward, as far as possible, by placing both hands on top of the other but without lifting their knees off the ground. Each participant performed the action twice, and the maximum height measurement was recorded to the nearest 0.1 cm. Systolic blood pressure (SBP) and diastolic blood pressure (DBP) were measured using an automatic blood pressure device (BPBio320s, Inbody Co., Seoul, Republic of Korea) after stabilizing the chair for approximately 10 min.

### 2.3. Data Analysis

Data were analyzed using IBM SPSS 26.0. Descriptive statistics were calculated to present the means, standard deviations, frequencies, and percentages of the measures. For the main analyses, first, a series of *t*-tests were used to determine whether there were any differences in the body compositions and physical fitness scores between the sexes. Second, a one-way analysis of variance was used to compare physical fitness across the disability levels and blood pressure levels according to gender. Then, post-hoc analysis using Scheffe’s test was carried out to determine which pairs of means were independently significant.

As a prerequisite step, Mahalanobis distances were calculated to examine multivariate outliers using a critical value of chi-square at *p* = 0.05, and cases deemed as multivariate outliers were excluded from the dataset. The quartile relative evaluation was established. A logistic regression analysis was performed to determine whether the participants’ fitness affected their blood pressure. The significance level was set at *p* < 0.05 for all tests.

## 3. Results

Table 1 shows differences in the participants’ physical attributes and disability types according to sex. BMI was not different between men and women, but among the fitness tests, the values for grip strength, sit-ups, PEI, and blood pressure (SBP, DBP) were higher for men than women; contrastingly, the values for the sit-and-reach tests were higher for women than men.

The differences in fitness according to disability level and blood pressure are presented by sex, i.e., male and female, in Table 2 and Table 3, respectively. Both males and females showed significant differences in grip strength according to disability level and blood pressure (*p* < 0.001), but there was no difference in grip strength according to blood pressure at disability Level 3 for females.

The sit-up test scores for males indicated significant differences according to disability level at all blood pressure levels and significant differences according to blood pressure level at disability levels 2 and 3 (*p* < 0.05). For females, the scores indicated the highest values according to disability level at normal blood pressure and elevated hypertension but indicated no difference in prehypertension and hypertension. There were no significant differences according to blood pressure at disability levels 1 and 2 for females.

The sit-and-reach test scores for both males and females indicated significant differences by disability level at all blood pressure levels. According to blood pressure level, both males and females showed no significant differences in all disability levels except disability Level 2 for males during the sit-and-reach test.

The cardiovascular endurance test was measured by PEI, and the higher the score, the better the cardiovascular endurance. The results of this study showed that there were significant differences in blood pressure levels at all disability levels for both males and females. However, according to disability level, the PEI scores for females showed no significant differences at all blood pressure levels except for elevated hypertension, while the PEI scores for males exhibited significant differences at all blood pressure levels except prehypertension.

There were significant differences in BMI according to blood pressure level at all disability levels for both males and females. However, according to disability level, there were no significant differences at all blood pressure levels except hypertension in males, but there were significant differences in BMI according to disability level at normal blood pressure and prehypertension for females.

Based on the odds ratios (OR) for blood pressure and physical fitness levels, the risk of elevated blood pressure, prehypertension, and hypertension were confirmed and are displayed in Table 4. Grip strength score increased risk by 1.78 times in Level 4 compared to Level 1 for elevated blood pressure, 3.26 times in Level 4 compared to Level 1 for prehypertension, and 3.5 times in Level 4 compared to Level 1 for hypertension. The muscular endurance test was measured using a sit-up test, and it exhibited an increase in risk by 1.30 times in Level 4 compared to Level 1 for elevated blood pressure and 1.70 times in Level 4 compared to Level 1 for hypertension but showed no significant difference in levels 2 and 3 compared to Level 1 for elevated blood pressure. Furthermore, there were no significant differences between levels 1 and 2 for prehypertension and hypertension according to all fitness levels.

The sit-and-reach test score, a flexibility indicator, indicated no significant differences between Levels 3 or 4 and Level 1 for elevated blood pressure, prehypertension, and hypertension but showed increased risk by 1.21, 1.23, and 1.30 times at Level 2 compared to Level 1 for elevated blood pressure, prehypertension, and hypertension. The PEI score, a cardiopulmonary endurance indicator, indicated that the risk increased by 0.63 in Level 4 compared to Level 1 for hypertension.

Finally, BMI showed an increase of 6.91 times in the obese group compared to the normal group for hypertension.

## 4. Discussion

This study investigated the relationship between physical fitness (including endurance, strength, and BMI) and blood pressure (different hypertension stages), considering differences in age, sex, and disability level among people with intellectual disabilities (IDs) to track the possible factors affecting changes in their physical function. Regarding BMI, males with intellectual disabilities showed higher levels than their female counterparts. These results compare with those of the National Health and Nutrition Examination Survey data in South Korea (2021), which showed that the obesity rate (BMI < 25) of individuals aged 19 and above without disabilities is 37.1%, with men showing a higher obesity rate (46.3%) compared with women (26.9%). However, in a study by Graham and Reid [23] targeting adults (aged 34–57 years), women with intellectual disabilities had both higher BMI and body fat percentages compared to their male counterparts. This finding is consistent with those of previous studies [20,21,24].

The results showed that blood pressure increased while grip strength decreased across all disability levels. Additionally, grip strength decreased as disability Level increased (up to Level 1). Furthermore, logistic regression analysis revealed that as the grip strength level decreased compared to normal grip strength, the risk of hypertension increased to a maximum risk of 3.5 times. In addition, based on the sit-ups test, the risk of hypertension increased to a maximum of 1.7 times as muscular endurance decreased. Walsh et al. [25] reported that blood pressure in individuals with intellectual disabilities is related to physical activity levels, and Oviedo et al. [26] stated that physical activity intervention plays an important role in physical fitness and the prevention of diseases and aging. Muscle strength and endurance are other independent risk factors for cardiovascular disease, metabolic syndrome, and cardiorespiratory fitness [26,27,28]. Therefore, as shown in the results of this study, it can be confirmed that low fitness levels and the risk of hypertension can be used as important health indicators for people with intellectual disabilities who lead sedentary lives, the data of which will be important for raising awareness about the significance of participating in physical activities to increase physical fitness levels. Especially for individuals with intellectual disabilities, physical activity effectively reduces the risk of chronic diseases and helps maintain a healthy weight by burning calories.

PEI showed a significant difference in cardiovascular endurance with increased blood pressure at all disability levels for both men and women. In particular, the change was more prominent in men than in women. Previous studies also reported that the cardiovascular endurance of men with intellectual disabilities is higher than that of women [20,23]. Furthermore, the results of logistic regression analysis showed that the lower the PEI grade, the higher the risk of hypertension.

Obesity is associated with adverse health effects related to metabolic syndrome and cardiovascular diseases. It also impacts physical fitness and exercise performance [29]. Individuals with intellectual disabilities are known to have higher obesity rates than those without disabilities. Therefore, increasing the physical activity and sports participation of individuals with intellectual disabilities is essential for managing their obesity. This study measured grip strength to assess muscular strength and used a sit-up test to evaluate muscular endurance, then determined the relationship between them and blood pressure, one of the CVD risk factors. Obesity and low physical activity are independent predictors of CVD and are major prophylactic factors for reducing CVD morbidity and mortality. These risk factors may be more significant for people with intellectual disabilities than for the general population because of the former’s low physical activity [30] and high levels of obesity [9]. In this study, it was confirmed that BMI increased as blood pressure increased at all disability levels for both males and females. In particular, for BMI indicating obesity, the risk of hypertension was up to 6.91 times higher. As with the general population in the previous study, the obesity of people with intellectual disabilities greatly increased the risk of hypertension, consistent with the results of this study. Obesity is a crucial risk factor for the premature mortality of people with disabilities [28]. Marín et al. [30] found that 36% of adults with Down syndrome were obese. A study of adult intellectual disabilities reported that the higher the BMI, the higher the obesity, and the higher the waist circumference, the higher the risk of hypertension. A study in Taiwan observed that 27.4% of adults with disabilities have hypertension [5].

This study was unable to conduct classifications other than Down syndrome due to the use of open data. Nonetheless, despite these limitations, this study represents the first paper to extensively investigate the correlation between physical fitness levels and blood pressure among individuals with intellectual disabilities. It is anticipated to be a significant research report as it directly describes the risk indicators of blood pressure and does so from a preventative perspective. In the future, systematic epidemiological studies should be conducted to further elucidate the relationship between blood pressure and physical fitness.

## 5. Conclusions

This study analyzed the physical fitness and blood pressure data assessment scores of people with intellectual disabilities and found differences by sex, age group, and disability levels. Significant differences in blood pressure according to fitness were confirmed, and significant effects on blood pressure risk were confirmed in terms of disability levels, obesity-related body composition, and physical fitness. However, this study, being a cross-sectional analysis based on data from open sources and focusing on physical fitness, is limited in its ability to control for missing data. Consequently, it was unable to confirm the effects of diseases, lifestyle, or exercise history. Moreover, despite the limitations of open data, this study holds significance in a context where research targeting individuals with disabilities is scarce and empirical data are lacking. Representing health and fitness-related data from South Korea, this paper has the potential to serve as a foundational resource for enhancing the generalizability of future studies.

Individuals with intellectual disabilities require social support. The findings of this study can be instrumental for policymakers in monitoring the physical health statuses of individuals with intellectual disabilities and devising appropriate activity and exercise programs. Engaging in regular sports and exercise can potentially assist individuals with intellectual disabilities in leading healthier lifestyles in the future.

## Figures and Tables

**Table 1 healthcare-12-00887-t001:** Difference in physical information and fitness by gender (*N* = 8502).

Factors		Male		Female			
		*N*	M ± SD/%	*N*	M ± SD/%	*t*	*p*
Disability type	ID	4951	95.5%	3279	98.9%	-	-
	Autism	234	4.5%	38	1.1%	-	-
Classification	1	2031	39.2%	1043	31.4%	-	-
	2	2167	41.8%	1554	46.8%	-	-
	3	987	19%	720	21.7%	-	-
Age (yr)	20–29	2778	53.6%	1466	44.2%	-	-
	30–39	1343	25.9%	829	25.0%	-	-
	40–49	689	13.3%	616	18.6%	-	-
	50–59	375	7.2%	406	12.2%	-	-
BMI (kg/m^2^)	18.5>/%	379	7.3%	237	7.10%	-	-
	18.5–22.9/%	1174	22.6%	752	22.7%	-	-
	23.0–24.9/%	750	14.5%	595	17.9%	-	-
	>25/%	2864	55.2%	1722	51.9%	-	-
Height (cm)		5175	167.8 ± 9.3	3308	153.9 ± 8.6	69.7	<0.001 ***
Weight (kg)		5175	73.2 ± 17.9	3308	61.4 ± 14.7	32.7	<0.001 ***
Grip strength (kg)		4964	23.4 ± 9.3	3146	16.9 ± 6.3	36.8	<0.001 ***
Sit up (times)		4567	20.4 ± 9.6	2602	14.6 ± 7.3	28.2	<0.001 ***
Sit-and-reach (cm)		4920	−3.9 ± 12.7	3197	2.7 ± 11.8	-24.1	<0.001 ***
PEI (score)		3810	46.2 ± 7.2	2440	38.9 ± 10.0	30.9	<0.001 ***
SBP (mmHg)		5106	124.1 ± 13.3	3270	119.7 ± 13.8	14.3	<0.001 ***
DBP (mmHg)		5106	79.9 ± 10.6	3270	77.9 ± 10.6	8.2	<0.001 ***

PEI (Physical Efficiency Index). SBP (Systolic Blood Pressure). DBP (Diastolic Blood Pressure). *** *p* < 0.001.

**Table 2 healthcare-12-00887-t002:** Difference in physical fitness by blood pressure and disability level in males with intellectual disabilities.

	Classification	Normal <120 (a)	Elevated 120–129 (b)	Prehypertension 130–139 (c)	Hypertension ≥140 (d)	*F*	*p*	Post Hoc
Grip strength (kg)	1	20.1 ± 9.1 (n = 666)	19.6 ± 8.2 (n = 588)	19.7 ± 8.4 (n = 352)	18.1 ± 7.2 (n = 226)	13.3	<0.001 ***	b, c, d < a
	2	25.7 ± 8.9 (n = 545)	25.6 ± 7.8 (n = 1006)	23.6 ± 8.4 (n = 323)	22.4 ± 7.9 (n = 226)	39.7	<0.001 ***	b, c, d < a
	3	33.3 ± 9.1 (n = 237)	31.4 ± 8.5 (n = 357)	29.6 ± 9.2 (n = 209)	28.9 ± 8.9 (n = 162)	27.7	<0.001 ***	b, c < a, d < b
	F	340.8	284.03	281.1	199.01			
	*p*	<0.001 ***	<0.001 ***	<0.001 ***	<0.001 ***			
	Post Hoc	3 > 2 > 1	3 > 2 > 1	3 > 2 > 1	3 > 2 > 1			
Sit up (times)	1	16.9 ± 8.6 (n = 606)	18.3 ± 9.8 (n = 531)	17.8 ± 8.3 (n = 308)	17.6 ± 8.6 (n = 200)	1.6	<0.093	-
	2	20.8 ± 8.5 (n = 520)	21.3 ± 9.0 (n = 966)	22.0 ± 9.7 (n = 296)	19.8 ± 8.6 (n = 206)	0.388	<0.011 *	d < c
	3	22.4 ± 8.8 (n = 226)	24.3 ± 11.4 (n = 328)	25.2 ± 9.0 (n = 188)	23.1 ± 9.7 (n = 141)	1.35	<0.025 *	-
	F	85.3	81.2	82.7	29.2			
	*p*	<0.001 ***	<0.001 ***	<0.001 ***	<0.001 ***			
	Post Hoc	3 > 1; 2 > 1	3 > 2 > 1	3 > 2 > 1	3 > 2, 1			
Sit-and-reach (cm)	1	−6.0 ± 13.8 (n = 683)	−6.5 ± 13.2 (n = 582)	−6.2 ± 13.3 (n = 355)	−5.8 ± 11.7 (n = 215)	0.002	<0.901	-
	2	−2.3 ± 11.3 (n = 550)	−4.4 ± 12.7 (n = 1004)	−3.6 ± 12.5 (n = 317)	−2.9 ± 13.6 (n = 231)	0.557	<0.015 *	b < a
	3	−0.67 ± 10.8 (n = 229)	−0.79 ± 11.8 (n = 345)	1.17 ± 10.7 (n = 196)	0.558 ± 11.8 (n = 148)	2.78	<0.188	-
	F	41.5	41.6	43.08	22.08			
	*p*	<0.001 ***	<0.001 ***	<0.001 ***	<0.001 ***			
	Post Hoc	3 > 1; 2 > 1	3 > 2 > 1	3 > 2 > 1	3 > 2, 1			
PEI (score)	1	46.4 ± 6.5 (n = 503)	44.7 ± 6.4 (n = 456)	44.7 ± 8.1 (n = 264)	41.9 ± 6.2 (n = 177)	26.6	<0.001 ***	b, c, d < a
	2	47.9 ± 7.2 (n = 426)	45.7 ± 6.0 (n = 822)	45.7 ± 7.2 (n = 240)	44.9 ± 7.2 (n = 192)	27.3	<0.001 ***	b, c, d < a
	3	48.7 ± 7.0 (n = 179)	46.9 ± 7.2 (n = 254)	45.6 ± 7.5 (n = 162)	44.4 ± 8.9 (n = 119)	55.8	<0.001 ***	b, c, d < a
	F	13.6	18.8	1.30	5.07			
	*p*	<0.001 ***	<0.001 ***	<0.376	<0.008 **			
	Post Hoc	3 > 1	3, 2 > 1	-	3, 2 > 1			
BMI (kg/m^2^)	1(	24.8 ± 5.2 (n = 137)	25.6 ± 5.4 (n = 398)	27.02 ± 5.6 (n = 249)	28.8 ± 6.7 (n = 1072)	45.5	<0.001 ***	c, d > a, b
	2	24.4 ± 5.0 (n = 150)	25.5 ± 5.0 (n = 501)	26.9 ± 5.5 (n = 342)	27.9 ± 5.2 (n = 1127)	20.24	<0.001 ***	c, d > a, b
	3	24.4 ± 4.2 (n = 45)	25.3 ± 4.6 (n = 226)	26.9 ± 4.5 (n = 131)	27.1 ± 4.6 (n = 572)	9.6	<0.001 ***	c, d > a, b
	F	1.80	0.755	0.012	9.05			
	*p*	<0.365	<0.665	<0.976	<0.011 *			
	Post Hoc	-	-	-	3 > 1			

Note: classification Level 1 (severe or profound) IQ and SQ of 34 or less, Level 2 (moderate) IQ and SQ of 35–49, Level 3 (mild) IQ and SQ of 50–70; IQ: Intelligence Quotient, SQ: Social Quotient. *** *p* < 0.001, ** *p* < 0.01, * *p* < 0.05.

**Table 3 healthcare-12-00887-t003:** Difference in physical fitness by hypertension and disability level in females with intellectual disabilities.

Factors	Classification	Normal <120 (a)	Elevated 120–129 (b)	Prehypertension 130–139 (c)	Hypertension ≥140 (d)	*F*	*p*	Post Hoc
Grip strength (kg)	1	14.9 ± 5.8 (n = 426)	15.4 ± 5.6 (n = 274)	14.5 ± 5.8 (n = 131)	13.06 ± 4.4 (n = 74)	23.06	<0.001 ***	b, c, d < a
	2	20.6 ± 6.3 (n = 522)	19.4 ± 5.0 (n = 712)	16.9 ± 6.1 (n = 143)	16.5 ± 5.5 (n = 120)	56.8	<0.001 ***	c > a, b; d > b
	3	20.8 ± 5.6 (n = 229)	20.4 ± 6.4 (n = 320)	20.4 ± 7.4 (n = 78)	19.09 ± 5.4 (n = 80)	5.17	<0.052	-
	F	221.2	128.2	45.5	33.8			
	*p*	<0.001 ***	<0.001 ***	<0.001 ***	<0.001 ***			
	Post Hoc	3 > 2 > 1	3 > 2 > 1	3 > 1; 2 > 1	3 > 1; 2 > 1			
Sit up (times)	1	12.8 ± 7.1 (n = 323)	13.7 ± 7.4 (n = 230)	13.9 ± 7.1 (n = 102)	12.4 ± 8.7 (n = 53)	0.355	<0.314	-
	2	15.01 ± 6.8 (n = 412)	14.5 ± 6.8 (n = 673)	14.6 ± 6.7 (n = 100)	14.6 ± 7.2 (n = 88)	0.520	<0.764	-
	3	15.6 ± 7.7 (n = 193)	17.2 ± 8.5 (n = 292)	14.6 ± 6.4 (n = 59)	15.3 ± 6.4 (n = 55)	0.234	<0.034*	-
	F	21.08	30.01	0.426	4.026			
	*p*	<0.001 ***	<0.001 ***	<0.759	<0.105			
	Post Hoc	3 > 1; 2 > 1	3 > 2 > 1	-	-			
Sit-and-reach (cm)	1	1.3 ± 13.6 (n = 447)	1.04 ± 11.8 (n = 298)	0.96 ± 11.9 (n = 131)	1.01 ± 9.8 (n = 76)	0.153	<0.977	-
	2	3.6 ± 11.3 (n = 518)	2.3 ± 11.6 (n = 733)	1.8 ± 10.8 (n = 141)	1.1 ± 12.5 (n = 121)	6.24	<0.080	-
	3	4.2 ± 12.7 (n = 224)	4.7 ± 10.8 (n = 315)	6.5 ± 10.1 (n = 76)	7.09 ± 11.3 (n = 77)	4.82	<0.160	-
	F	9.5	16.2	10.5	11.3			
	*p*	<0.005 **	<0.001 ***	<0.002 **	<0.001 ***			
	Post Hoc	3 > 1; 2 > 1	3 > 2, 1	3 > 2, 1	3 > 2, 1			
PEI (score)	1	40.0 ± 10.8 (n = 314)	36.7 ± 7.6 (n = 212)	36.5 ± 9.3 (n = 106)	36.9 ± 10.1 (n = 59)	2.6	<0.003 **	b < a
	2	41.1 ± 10.6 (n = 405)	36.8 ± 7.4 (n = 568)	40.9 ± 12.6 (n = 115)	37.9 ± 12.0 (n = 95)	10.9	<0.001 ***	b, d < a; c < b
	3	41.6 ± 10.2 (n = 178)	38.5 ± 9.3 (n = 249)	40.9 ± 12.6 (n = 68)	38.8 ± 13.2 (n = 61)	8.2	<0.001 ***	b, d < a
	F	2.55	5.34	5.17	0.796			
	*p*	<0.258	<0.018 *	<0.051	<0.673			
	Post Hoc	-	2 > 1	-	-			
BMI (kg/m^2^)	1	24.3 ± 4.8 (n = 77)	25.9 ± 5.7 (n = 201)	26.3 ± 5.4 (n = 194)	28.2 ± 6.3 (n = 448)	12.7	<0.001 ***	b, c, d > a
	2	25.4 ± 5.4 (n = 71)	25.8 ± 5.0 (n = 351)	28.4 ± 5.6 (n = 245)	29.2 ± 5.5 (n = 839)	35.2	<0.001 ***	c, d > a, b
	3	24.7 ± 5.7 (n = 54)	25.4 ± 4.4 (n = 156)	27.3 ± 6.4 (n = 129)	27.3 ± 5.3 (n = 373)	8.5	<0.003 **	c, d > a, b
	F	2.44	1.34	2.55	0.909			
	*p*	<0.003 **	<0.396	<0.011 *	<0.065			
	Post Hoc	2 > 1	-	2 > 1	-			

Note: classification Level 1 (serve or profound) IQ and SQ of 34 or less, Level 2 (moderate) IQ and SQ of 35–49, Level 3 (mild) IQ and SQ of 50–70; IQ: Intelligence Quotient, SQ: Social Quotient. *** *p* < 0.001, ** *p* < 0.01, * *p* < 0.05.

**Table 4 healthcare-12-00887-t004:** Odds ratio (95% confidence interval) for blood pressure and fitness levels compared to baseline levels.

Fitness Factors		Normal <120	Elevated 120–129	Prehypertension 130–139	Hypertension ≥140
			Odds ratio (95% confidence interval) and *p*		
Grip strength(kg)	1st (n = 2030)	Reference			
	2nd (n = 2007)		0.99 (0.86–1.13) <0.891	1.24 (1.02–1.52) <0.032 *	1.18 (0.93–1.49) <0.153
	3rd (n = 1993)		1.30 (1.12–1.49) <0.001 ***	1.78 (1.46–2.17) <0.001 ***	1.6 (1.27–2.01) <0.001 ***
	4th (n = 1976)		1.78 (1.53-2.07) <0.001 ***	3.26 (2.67–3.98) <0.001 ***	3.5 (2.87–4.46) <0.001 ***
Sit-ups (times)	1st (n = 1949)	Reference			
	2nd (n = 1915)		1.06 (0.91–1.22) <0.409	1.16 (0.95–1.42) <0.142	1.07 (0.85–1.34) <0.535
	3rd (n = 1464)		1.12 (0.96–1.32) <0.134	1.43 (1.15–1.77) <0.001 ***	1.26 (0.99–1.59) <0.056
	4th (n = 1768)		1.30 (1.12–1.52) <0.001 ***	1.70 (1.39–2.09) <0.001 ***	1.23 (0.97–1.55) <0.079
Sit-and-reach (cm)	1st (n = 2095)	Reference			
	2nd (n = 1948)		1.21 (1.05–1.4) <0.009 **	1.23 (1.02–1.49) <0.029 *	1.30 (1.04–1.61) <0.019 *
	3rd (n = 1971)		0.96 (0.83–1.11) <0.606	1.05 (0.87–1.27) <0.554	1.18 (0.99–1.46) <0.124
	4th (n = 1998)		0.95 (0.82–1.09) <0.495	0.86 (0.71–1.04) <0.128	1.01 (0.814–1.25) <0.915
PEI (score)	1st (n = 1572)	Reference			
	2nd (n = 1540)		0.50 (0.42–0.59) <0.001 ***	0.71 (0.57–0.89) <0.003 **	0.42 (0.33–0.54) <0.001 ***
	3rd (n = 1562)		0.76 (0.64–0.90) <0.001 ***	0.88 (0.70–1.10) <0.286	0.47 (0.37–.611) <0.001 ***
	4th (n = 1550)		0.76 (0.64–0.90) <0.002 **	0.88 (0.75–1.10) <0.280	0.63 (0.50–0.80) <0.001 ***
BMI (kg/m^2^)	Normal (n = 605)	Reference			
	Underweight (n = 1902)		1.18 (0.96–1.43) <0.104	1.27 (0.93–1.74) <0.128	2.1 (1.38–3.54) <0.001 ***
	Overweight (n = 1319)		1.40 (1.13–1.74) <0.002 **	1.78 (1.29–2.46) <0.001 ***	3.86 (2.38–6.26) <0.001 ***
	Obesity (n = 4524)		1.76 (1.46–2.12) <0.001 ***	2.97 (2.20–3.90) <0.001 ***	6.91 (4.39–10.87) <0.001 ***

*** *p* < 0.001, ** *p* < 0.01, * *p* < 0.05.

## Data Availability

Data are contained within the article.

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
