# Peer review of "A Study of Blood Pressure and Physical Fitness in People with Intellectual Disabilities in South Korea"

_healthcare, 2024, doi:10.3390/healthcare12090887_

Round 1
Reviewer 1 Report
Comments and Suggestions for Authors
Thank you for the opportunity to review this article. The author aimed to examine the relationship between physical fitness and blood pressure risk, in people with intellectual disabilities. I would like to congratulate the author, the topic of the article is interesting and I particularly appreciate the focus not only on the effectiveness of physical activity but especially on the protective effect of different levels of fitness.
I think the article needs some improvements, but then it can be published.
Here are my suggestions:
· Abstract (line 9): To avoid repetition, I suggest replacing “individuals without disabilities” with “healthy individuals”
· Abstract (line 10): it is not clear what the author means by “factors”, I invite him to rephrase the sentence.
· Abstract (line 12): I invite the author to replace the term “sex” with “gender” throughout the article.
· Abstract (lines 17-18): I invite the author to make abbreviations explicit before using them in the text.
· Introduction (line 40): For the sake of clarity, I suggest that the author sorts the age groups in ascending order.
· Introduction (line 62): I think it would be useful to add the bibliographic reference to the above-mentioned data before the point.
· Materials and Methods (lines 86-87): I suggest the author remove the “.0%”
· Materials and Methods (table 1): I suggest the author to insert the units of the various measurements taken, and to make the abbreviations explicit
· Materials and Methods (line 92): The title of the paragraph refers to BMI Measurement, but nothing is written about it in the text. I invite the author to correct this omission
· Materials and Methods (line 139): In the description of the sit-and-reach test, the author states that the subjects had to place one hand on top of the box for measurement. Instead, the correct procedure requires the subject to place both hands. I invite the author to clarify this point.
· Results (line 171): In the title of Table 2, I suggest the author replace “hypertension” with “blood pressure”.
· Results (line 174): typo, “serve” instead of “severe”
· Results (lines 186-187): there is a repetition of the term score/scores
· Discussion (lines 233-235): the author claims that 'Men with intellectual disabilities were taller and heavier than women with intellectual disabilities', I think this is a foregone statement, I suggest removing it.
· Discussion (lines 249-251): The connection of this period with the preceding text is not clear.
Author Response
# Reviewer 1
- Abstract (line 9): To avoid repetition, I suggest replacing “individuals without disabilities” with “healthy individuals”
A: I edited the words based on your comment.
- Abstract (line 10): it is not clear what the author means by “factors”, I invite him to rephrase the sentence.
A: I changed the words in the sentence to make it easier to understand.
- Abstract (line 12): I invite the author to replace the term “sex” with “gender” throughout the article.
A: I changed the term with ‘gender’.
- Abstract (lines 17-18): I invite the author to make abbreviations explicit before using them in the text.
A: I wrote the abbreviations to understand easily.
- Introduction (line 40): For the sake of clarity, I suggest that the author sorts the age groups in ascending order.
A: I changed the ascending order.
- Introduction (line 62): I think it would be useful to add the bibliographic reference to the above-mentioned data before the point.
A: I add the information of the reference as follows: These findings confirm that prior research has established a correlation between health-related physical fitness and CVD in individuals with intellectual disabilities
- Materials and Methods (lines 86-87): I suggest the author remove the “.0%”
A:. Thank you. 0% was deleted.
- Materials and Methods (table 1): I suggest the author to insert the units of the various measurements taken, and to make the abbreviations explicit
A: I added units and abbreviations
- Materials and Methods (line 92): The title of the paragraph refers to BMI Measurement, but nothing is written about it in the text. I invite the author to correct this omission
A: I added measurement of BMI as follows: The BMI was calculated by dividing body weight (kg) by height in square meters (m2).
- Materials and Methods (line 139): In the description of the sit-and-reach test, the author states that the subjects had to place one hand on top of the box for measurement. Instead, the correct procedure requires the subject to place both hands. I invite the author to clarify this point.
A: Based on your feedback, I've checked it out and changed it to two hand.
- Results (line 171): In the title of Table 2, I suggest the author replace “hypertension” with “blood pressure”.
A: I replaced “hypertension” with “blood pressure”.
- Results (line 174): typo, “serve” instead of “severe”
A: I edit the term. Than you for the comment.
- Results (lines 186-187): there is a repetition of the term score/scores
A: I changed the word to ‘values’ instead of score.
- Discussion (lines 233-235): the author claims that 'Men with intellectual disabilities were taller and heavier than women with intellectual disabilities', I think this is a foregone statement, I suggest removing it.
A: I appreciate your nice comment. I removed the sentence.
- Discussion (lines 249-251): The connection of this period with the preceding text is not clear.
A: Based on your feedback, I've changed the location of this paragraph to make the content more connected (line 267)
Reviewer 2 Report
Comments and Suggestions for Authors
The author conducted a cross-sectional study to evaluate the association between blood pressure and physical function in adults with intellectual disabilities leveraging NFSC survey data (2018-2021). The author reports that lower grip strength, lower PEI scores, and higher BMI were associated with increased blood pressure and higher le els of disability were associated with increased blood pressure. The research question is of public health importance. Suggestions:
1) Please add a STROBE checklist in the appendix.
2) Abstract: (i) Please spell out PEI in the abstract. (ii) Please provide some numerical results in the results section of the abstract.
3) Introduction: please add some information about existing literature related to this topic (e.g. PMID: 38238381, PMID: 30276179, PMID: 32059987) and please justify why answering this research question is important.
4) Methods: In the first paragraph, please describe inclusion/exclusion criteria and how you arrived at the final analytic sample... i.e. please provide number of total survey respondents you started with and number excluded for each exclusion criteria.
5) What is the survey response rate for NFSC? Were there any weights used for sampling techniques or to address response bias?
6) Please consider making a multivariable adjusted logistic regression model to adjust for potential confounders.
7) For bivariate analyses, are p-values adjusted for multiple comparisons testing? e.g. Bonferroni correction?
8) What was the quantity of missing data? How was missing data handled or accounted for in the analyses?
9) In tables 2, 3, and 4 please add frequencies (total number of respondents) for each row and each column.
10) In the discussion (around lines 265-266) where increasing physical activity levels is mentioned, please expand the discussion by adding information about benefits of increasing physical activity levels (e.g. PMID: 28125943, PMID: 37361407, PMID: 38129104)
11) Please add a paragraph discussing limitations of the study as a last para in the discussion section, just before conclusion section. Discuss limitations of this study; for e.g. response bias, selection bias, potential for misclassification of the primary variables considered and chance of measurement error, unmeasured and residual confounding, temporality issues etc and how these issues might bias the results (direction of bias).
12) Please verify references; some of them appear to be numbered incorrectly; e.g. line 283 - "Marín et al. [29]" is actually reference number [22] in your reference list.
Comments on the Quality of English Languagen/a
Author Response
# Reviewer 2
1) Please add a STROBE checklist in the appendix.
A: This study was conducted in accordance with the STROBE (Strengthening the Reporting of Observational Studies in Epidemiology) checklist to ensure rigorous and transparent reporting of results.
2) Abstract: (i) Please spell out PEI in the abstract. (ii) Please provide some numerical results in the results section of the abstract.
A: I put an abbreviation for PEI. But if I include some numerical results, I consider that it would be a problem with the character limit for the summary.
3) Introduction: please add some information about existing literature related to this topic (e.g. PMID: 38238381, PMID: 30276179, PMID: 32059987) and please justify why answering this research question is important.
A: Following your feedback, I added some information: Numerous studies have indicated that individuals with intellectual disabilities engage in less physical activity compared to healthy individuals, leading to a higher prevalence of metabolic syndrome, including hypertension and diabetes. These studies have particularly emphasized the relationship between physical fitness and health outcomes. Among the components of physical fitness, cardiorespiratory endurance, muscular strength and endurance, and obesity have been identified as factors contributing to the increased risk of hypertension [11, 12, 13].
4) Methods: In the first paragraph, please describe inclusion/exclusion criteria and how you arrived at the final analytic sample... i.e. please provide number of total survey respondents you started with and number excluded for each exclusion criteria.
A: We used open source data that there is no specific inclusion/exclusion criteria.
5) What is the survey response rate for NFSC? Were there any weights used for sampling techniques or to address response bias?
A: Since this current article uses open data sources, we do not know information about response rates or response bias.
6) Please consider making a multivariable-adjusted logistic regression model to adjust for potential confounders.
A: Logistic regression was used because the study was looking at how each measure affected how hypertension stages changed
7) For bivariate analyses, are p-values adjusted for multiple comparisons testing? e.g. Bonferroni correction?
A: Bonferroni correction was not performed in this study and multivariate analysis was performed without adjustment for age and sex.
8) What was the quantity of missing data? How was missing data handled or accounted for in the analyses?
A: Although not specified separately, the number of participants can be determined through Table 1.
9) In tables 2, 3, and 4 please add frequencies (total number of respondents) for each row and each column.
A: Writing out each value would complicate the table excessively, making it difficult to include. Nevertheless, if you still wish to see the values, please request again, and I will incorporate them accordingly.
10) In the discussion (around lines 265-266) where increasing physical activity levels is mentioned, please expand the discussion by adding information about benefits of increasing physical activity levels (e.g. PMID: 28125943, PMID: 37361407, PMID: 38129104)
A: We put the benefits of increasing physical activity levels based on your feedback as follows: Especially for individuals with intellectual disabilities, physical activity effectively reduces the risk of chronic diseases and helps maintain a healthy weight by burning calories.
11) Please add a paragraph discussing limitations of the study as a last para in the discussion section, just before conclusion section. Discuss limitations of this study; for e.g. response bias, selection bias, potential for misclassification of the primary variables considered and chance of measurement error, unmeasured and residual confounding, temporality issues etc and how these issues might bias the results (direction of bias).
A: The limitations of this study were discussed on line 303-310. The paragraph was modified for readability. And I added the limitations of the study regarding response and selection bias.
12) Please verify references; some of them appear to be numbered incorrectly; e.g. line 283 - "Marín et al. [29]" is actually reference number [22] in your reference list.
A: Thank you for your comment. I changed reference number.
Round 2
Reviewer 2 Report
Comments and Suggestions for Authors
Thank you for submitting a revised manuscript. I only have minor comments...Comments 1, 8, and 9 from round 1 review haven't been addressed. Quantity of missing data typically varies between variables, i.e. for one variable, data might be available for 100 participants whereas for another variable, data might be available for 200 participants. As per STROBE guidelines, describing handling of missing data is important. Please complete a STROBE checklist and attach it as an appendix; and please address comments 8 and 9 from round 1 review.
Comments on the Quality of English Languagen/a
Author Response
Reviewer 2
1) Please add a STROBE checklist in the appendix.
I added a checklist in the appendix.
8) What was the quantity of missing data? How was missing data handled or accounted for in the analyses?
As this study utilized open data, it was not possible to add missing data. Instead, this limitation was addressed in the conclusion section. Additionally, the number of participants was included in the table 2,3,4 to provide clarity on the sample size.
Edited limitation in conclusion is as follows:
However, this study, being a cross-sectional analysis based on data from open sources and focusing on physical fitness, is limited in its ability to control for missing data. Consequently, it was unable to confirm the effects of diseases, lifestyle, or exercise history. Moreover, despite the limitations of open data, this study holds significance in a context where research targeting individuals with disabilities is scarce, and empirical data is lacking. Representing health and fitness-related data from South Korea, this paper has the potential to serve as a foundational resource for enhancing the generalizability of future studies.
9) In tables 2, 3, and 4 please add frequencies (total number of respondents) for each row and each column.: Table 2,3,,4
I added frequencies based on your feedback.